# DAF-2c signaling promotes taste avoidance after starvation in *Caenorhabditis elegans* by controlling distinct phospholipase C isozymes

Masahiro Tomioka [1✉], Moon Sun Jang[1,2] & Yuichi Iino [1]

Previously, we reported that DAF-2c, an axonal insulin receptor isoform in *Caenorhabditis elegans*, acts in the ASER gustatory neuron to regulate taste avoidance learning, a process in which worms learn to avoid salt concentrations experienced during starvation. Here, we show that secretion of INS-1, an insulin-like peptide, after starvation conditioning is sufficient to drive taste avoidance via DAF-2c signaling. Starvation conditioning enhances the salt-triggered activity of AIA neurons, the main sites of INS-1 release, which potentially promotes feedback signaling to ASER to maintain DAF-2c activity during taste avoidance. Genetic studies suggest that DAF-2c–Akt signaling promotes high-salt avoidance via a decrease in PLCβ activity. On the other hand, the DAF-2c pathway promotes low-salt avoidance via PLCε and putative Akt phosphorylation sites on PLCε are essential for taste avoidance. Our findings imply that animals disperse from the location at which they experience starvation by controlling distinct PLC isozymes via DAF-2c.

[1] Department of Biological Sciences, Graduate School of Science, The University of Tokyo, Bunkyo-ku, Tokyo 113-0033, Japan. [2]Present address: Neuroscience Institute, Graduate School of Science, Nagoya University, Nagoya, Aichi 464-8602, Japan. ✉email: tomioka@g.ecc.u-tokyo.ac.jp

Neuropeptide signaling regulates a variety of neuronal functions to increase the behavioral repertoire of animals, thereby affecting behaviors such as foraging and escape[1–3]. Insulin-like peptides (ILPs), including insulin, IGF1, and IGF2, also function in behaviors such as those related to learning. For example, IGF1 peptides expressed in mitral cells in the olfactory bulb and midbrain dopamine neurons promote social learning[4] and motor skill learning[5], respectively. IGF2 expression in the hippocampus enhances memory retention[6]. Recent advances in genetic and fluorescence imaging techniques have helped elucidate the neuropeptide signaling that acts at specific sites and timing within neural circuits[7,8]. However, our understanding of these processes during behavior and behavioral learning remains limited.

In *Caenorhabditis elegans*, an ILP, INS-1, acts on an insulin receptor homolog, DAF-2, and its downstream PI 3-kinase (PI3K)–Akt signaling in the ASER gustatory neuron, the right-sided ASE class of amphid sensory neurons, to regulate taste avoidance learning, a process in which worms learn to avoid salt concentrations experienced during starvation[9,10]. The *daf-2* gene produces DAF-2 (E11.5−) isoforms, e.g., DAF-2a, and DAF-2 (E11.5+) isoforms, e.g., DAF-2c, by alternative splicing of exon 11.5[11]. In ASER, DAF-2a mainly localizes to the cell body to regulate starvation-dependent nuclear translocation of the FOXO homolog, DAF-16, whereas DAF-2c is preferentially translocated to the axon, and its translocation increases by starvation[10]. DAF-2a and DAF-2c are required for taste avoidance learning in the cell body and axon of ASER, respectively[10,12]. In addition to taste avoidance learning, INS-1 and other ILPs regulate various forms of behavioral learning in C. elegans, such as those related to olfaction, thermosensation, and pathogen avoidance[13–20]. In the mammalian brain, the insulin receptor isoforms IR-A and IR-B, which are produced in a manner similar to that observed in *C. elegans*, are also coexpressed in neurons[21], although their functions are currently unclear.

Diacylglycerol (DAG) metabolism plays crucial roles in the regulation of behavioral learning in sensory neurons, such as the thermosensory AFD, the olfactory AWC, and the gustatory ASE neurons[22–25]. In the axon of ASER, EGL-8 (phospholipase Cβ, PLCβ) is proposed to regulate synaptic output via DAG and PKC-1 (novel protein kinase Cε/η) localized at presynaptic regions[26]. Correlations exist between DAG dynamics and migration direction in salt chemotaxis of worms after feeding: DAG levels at the ASER axon are increased by a decrease in ambient salt concentration, whereby well-fed worms migrate toward high salt concentrations; in contrast, DAG levels are decreased by an increase in ambient salt concentration, thereby leading worms to move toward low salt concentrations. This mechanism allows worms to remain in the location at which they were fed[26]. PLC-1 (an isozyme of the PLC family, PLCε) plays a minor role in DAG dynamics in the ASER axon of well-fed worms[26]. PLC-1 is proposed to regulate neurotransmission from ASER to its downstream interneuron, AIB[25]. In mammals, PLCε is activated by direct interaction with small GTPases and mediates signaling pathways, such as Ca$^{2+}$ signaling, in several cell types[27]. *C. elegans* PLC-1 harbors the domains required for binding such GTPases[28], although it is unclear how PLC-1 activity is regulated.

Here, we show that INS-1 secretion only during salt chemotaxis after starvation conditioning restores defects in taste avoidance learning of *ins-1* mutants. These effects depend on DAF-2c but not DAF-16. Therefore, INS-1–DAF-2c signaling during salt chemotaxis after starvation conditioning is important for taste avoidance learning. The responses of AIA interneurons, which are downstream of ASER in the neural circuit and a main site of INS-1 release, to changes in salt concentration are elevated after starvation conditioning dependent on the synaptic output of

ASER. Based on these findings, we propose that INS-1 release from AIA is enhanced during chemotaxis, where worms undergo multiple changes in salt concentration, after starvation conditioning. This feedback signal of INS-1 to DAF-2c in ASER may contribute to long-lasting activation of DAF-2c signaling to induce the robust avoidance behavior. In addition, ASER-specific knockdown via a CRISPR/Cas9 strategy suggests that decreased EGL-8 activity in ASER suppresses the aberrant high-salt migration in the *daf-2c* mutant after starvation conditioning. Further genetic studies suggest that DAF-2c signaling promotes high-salt migration via PLC-1 phosphorylation after starvation conditioning. Based on these findings, we propose a model in which DAF-2c signaling modulates the balance of pathways mediated by PLC isozymes in ASER to direct dramatic behavioral changes after starvation.

## Results

### INS-1 acts on the DAF-2c pathway for both low- and high-salt migration after starvation.

We have reported that worms with a large deletion mutation of *ins-1*, namely *ins-1(nr2091)*[29], were attracted to salt concentrations experienced during feeding similar to the wild type (Supplementary Fig. 1); however, they showed significant defects in both low- and high-salt migration after high- and low-salt conditioning in the absence of food, respectively (Fig. 1a, b)[25]. *daf-2c(pe2722)* mutants, which harbor a frameshift deletion in the cassette exon 11.5, and *daf-16(mgDf50)* mutants, which harbor a large deletion in a *daf-16* locus, also show significant defects in taste avoidance learning (Fig. 1b)[12]. On the other hand, the *daf-2c* and *daf-16* mutants show strong attraction to salt concentrations experienced during feeding, although the *daf-2c* mutant shows weaker attraction toward low salt than that in the wild type (Supplementary Fig. 1)[12]. The *daf-16* and *daf-2c* mutations show additive effects on taste avoidance learning and the *daf-16; daf-2c* double-mutants show aberrant salt chemotaxis, as if they were attracted to salt concentrations experienced during starvation (Fig. 1b)[12]. To assess which pathway acts downstream of INS-1, the effects of *ins-1(nr2091)* on taste avoidance learning were examined using single or double *daf-2c(pe2722)* and *daf-16(mgDf50)* mutants. The *ins-1* mutation enhanced the defects of *daf-16* but not *daf-2c* mutants in low-salt migration after high-salt conditioning, suggesting that *ins-1* acts in the *daf-2c* genetic pathway (Fig. 1b left, c). The *ins-1* mutation enhanced the defects of both *daf-2c* and *daf-16* mutants in high-salt migration after low-salt conditioning (Fig. 1b, right). On the other hand, it did not further enhance the salt chemotaxis defect of the *daf-16; daf-2c* double-mutant after starvation conditioning at low salt (Fig. 1b, right). Therefore, *ins-1* acts on both the *daf-2c* and *daf-16* genetic pathways to promote high-salt migration after starvation conditioning at low salt (Fig. 1c).

### INS-1 release during salt chemotaxis drives taste avoidance.

As previously reported, *ins-1* mutants show a significant defect in learned taste avoidance after starvation conditioning in liquids and this defect is rescued by INS-1 expression in neurons including AIA[9]. The learning defect of *ins-1* was also rescued by INS-1 expression in either ASI or AWA, but not in ASJ, RIA or ADF, in which intrinsic expression of *ins-1* was observed by using a fosmid reporter (Supplementary Fig. 2a–c). We also confirmed the requirement of INS-1 in AIA and chemosensory neurons, including ASI, for taste avoidance learning after starvation conditioning on agar plates by conditional knockdown of *ins-1* using the somatic CRISPR/Cas9 method[30] (Supplementary Fig. 3).

Although INS-1 expression has not been confirmed in RIC interneurons, ectopic expression of INS-1 in RIC largely restored the learning defects of *ins-1*, which suggests that INS-1 could

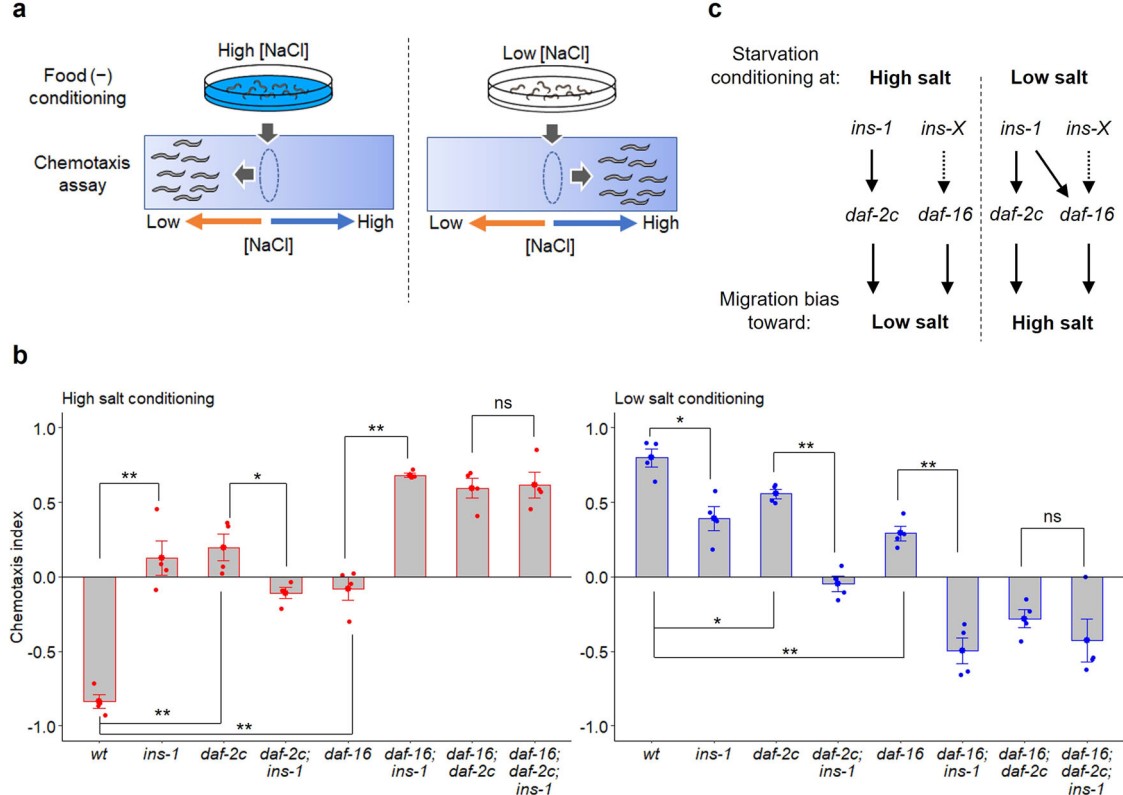

**Fig. 1 INS-1 acts in both DAF-2c and DAF-16 pathways in taste avoidance learning. a** Schematic of taste avoidance learning. **b** Salt chemotaxis after conditioning on agar plates at high or low salt concentrations in the absence of food. A chemotaxis index was determined according to the following equation: Chemotaxis index $= (N_A - N_B)/(N_{all} - N_C)$, where $N_A$ and $N_B$ are the number of worms in the high- and low-salt areas, respectively, $N_{all}$ is the total number of worms on a test plate, and $N_C$ is the number of worms in the area around the starting position. Positive and negative chemotaxis indices mean migration toward high and low salt, respectively. Each dot in red or blue represents a chemotaxis index calculated in each chemotaxis assay after conditioning at high or low salt concentrations, respectively. $n = 4$ assays. Bars represent mean values; error bars represent SEM. Two-tailed Welch's t-test with Holm correction: $*P < 0.05$ and $**P < 0.01$. ns not significant. **c** Proposed genetic pathways downstream of $ins-1$ in the regulation of low- or high-salt migration after high- or low-salt conditioning in the absence of food, respectively.

function extrasynaptically in taste avoidance learning (Supplementary Fig. 2b, 2nd block from right). RIC-specific knockdown of $ins-1$ caused a weak but significant defect after starvation conditioning only at low salt, implying that endogenous INS-1 in RIC may weakly contribute to the regulation of taste avoidance learning (Supplementary Fig. 3). It was reported that the action of RIC interneurons is repressed by dopaminergic signaling[31]. Thus, we examined the effect of excess dopamine release by a deletion mutation of $dat-1$, the dopamine reuptake transporter, on taste avoidance learning of the $ins-1$ mutant expressing INS-1 only in RIC. Indeed, the $dat-1(ok157)$ mutation inhibited the effect of INS-1 expression in RIC on taste avoidance learning of the $ins-1$ mutant (Supplementary Fig. 2e). This effect was not observed in $cat-2$ mutants, which are defective in dopamine synthesis, or $dop-3$ mutants, which carry a deletion mutation in a DOP-2-like receptor acting in RIC; this confirms that excess dopamine signaling repressed the action of INS-1 in RIC (Supplementary Fig. 2e).

We next sought to determine the timing of INS-1 action in taste avoidance learning. We generated $ins-1$ mutants in which INS-1 was expressed in RIC neurons and VR1, a capsaicin receptor, was expressed in dopaminergic neurons (Fig. 2a). Then, activities of the dopaminergic neurons were increased by capsaicin application at different periods, so that the action of INS-1 expressed in RIC was repressed by increased dopamine signaling[31] (Fig. 2b). Capsaicin application only during chemotaxis assay but not during starvation conditioning, inhibited the rescue effects of INS-1 expression in RIC on the learning defect of $ins-1$ mutants (Fig. 2c, fourth blocks, red

bars). This capsaicin-induced inhibitory effect was not observed in the $cat-2$ mutants (Fig. 2c, fifth blocks, red bars); this confirms that excess dopamine signaling repressed the action of INS-1 in RIC. These data suggest that INS-1 secretion during salt chemotaxis but not starvation conditioning is required for taste avoidance learning. We note that capsaicin application during both conditioning and chemotaxis periods caused a low-salt migration defect after high-salt conditioning in the $cat-2$ mutant background (Fig. 2c left, the fifth block, the purple bar), which might be an unknown effect of prolonged capsaicin exposure on salt chemotaxis.

To further examine timing-dependent action of INS-1, INS-1 and VR1 were coexpressed in the dopaminergic neurons and capsaicin was applied to activate the dopaminergic neurons and promote INS-1 release at different periods in the $ins-1$ mutants (Fig. 3a, b)[32]. First, we confirmed that the learning defect of the $ins-1$ mutants was not rescued by either INS-1 expression or activation of the dopaminergic neurons alone (Fig. 3c, 2nd and 3rd blocks). We note that capsaicin application during conditioning and/or chemotaxis periods caused decreased high-salt migration in worms expressing VR1 alone (Fig. 3c, third blocks), implying that excess dopamine release might affect salt chemotaxis in a direction that decreases migration toward high salt. On the other hand, the learning defect of the $ins-1$ mutants was rescued by INS-1 expression combined with capsaicin application during salt chemotaxis after starvation conditioning: INS-1 release during salt chemotaxis promoted movement toward both lower and higher salt concentrations dependent on

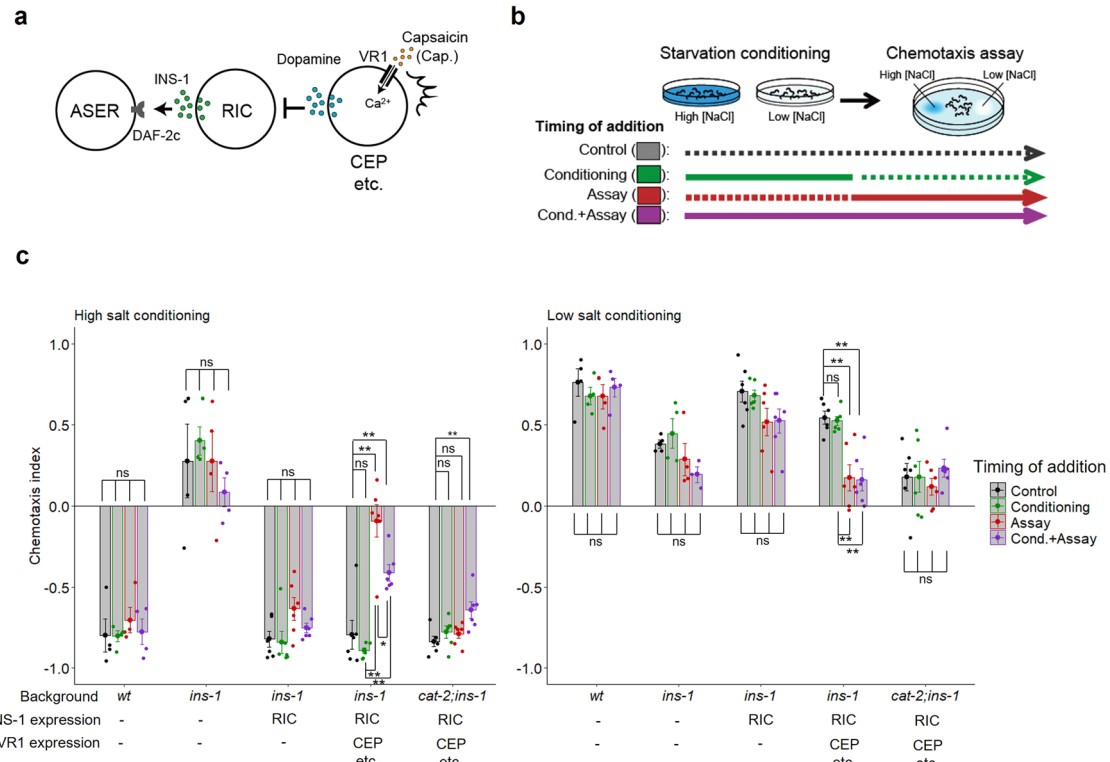

**Fig. 2 INS-1 release during salt chemotaxis but not during starvation conditioning is required for taste avoidance learning. a** INS-1::Venus and VR1 were expressed in RIC under the *tbh-1* promoter and in dopaminergic neurons, including CEP, under the *dat-1* promoter, respectively, and then capsaicin was applied to inhibit INS-1 release from RIC with specific timing. **b** Timing of capsaicin addition during the taste avoidance learning paradigm. Continuous lines and dotted lines denote with and without capsaicin, respectively. **c** Chemotaxis of wild type, *ins-1(nr2091)* or *cat-2(e1112); ins-1(nr2091)* mutant worms with or without transgenes that drive expression of INS-1::Venus or both INS-1::Venus and VR1. Starvation conditioning was performed at high (left) or low (right) salt concentrations. Capsaicin was applied only during salt conditioning (green dots), only during chemotaxis test (red dots), or during both (purple dots). $n = 4–6$ assays. Bars represent mean values; error bars represent SEM. ANOVA with Tukey's post hoc test: $**P < 0.01$. ns not significant.

conditioned salt concentrations (Fig. 3c, fourth blocks, red bars). These effects were observed in the *daf-16* but not *daf-2c* mutant (Fig. 3d). Hence, the data suggest that INS-1 drives dispersal from salt concentrations encountered by worms under starvation conditions via DAF-2c. On the other hand, forced INS-1 release only during starvation conditioning enhanced (rather than rescued) the learning defect of the *ins-1* mutant (Fig. 3c, fourth blocks, green bars), implying that excess amount of INS-1 release during starvation conditioning may reduce learned taste avoidance by an unknown mechanism. Consistent of this, the inhibition of INS-1 release only during a chemotaxis period had a higher inhibitory effect on taste avoidance than that during both conditioning and chemotaxis (Fig. 2c, left, the fourth block, red vs. purple bars). Collectively, our results indicate that the timing of INS-1 release is important for the regulation of taste avoidance learning, and INS-1 release during salt chemotaxis is required and sufficient for taste avoidance.

**Salt responses of AIA neurons are increased by starvation depending on synaptic output from ASER.** It has been reported that spontaneous activity of the AIA neuron, a main site of INS-1 release, is increased during foraging after food deprivation for more than 20 min[33]. Here, we examined the activity of AIA neurons in response to salt concentration changes using $Ca^{2+}$ imaging, making comparisons following conditioning in the presence or absence of food. We first monitored AIA activities following salt concentration changes between 50 and 25 mM. As previously reported[34], after cultivation with 50 mM (high) salt in the presence of food, AIA neurons were activated in response to increased salt concentration,

whereas strong activation did not occur in response to decreased salt concentration (Fig. 4a left, blue line in top panel, middle panel). After food deprivation for 5–6 h with high salt, the OFF response, i.e., calcium response to a decrease in salt concentration, was significantly enhanced (Fig. 4a left, orange line in top panel, bottom panel, 4b left). We next assessed AIA responses after cultivation at 25 mM (low) salt in the presence of food. The AIA neurons were activated in response to both decreased and increased salt concentrations, with stronger activation observed in OFF responses (Fig. 4a right, blue line in top panel, middle panel). After food deprivation for 5–6 h with low salt, the ON response, i.e., calcium response to an increase in salt concentration, was significantly enhanced (Fig. 4a right, orange line in top panel, bottom panel, Fig. 4b right). To test the contribution of ASER to these AIA responses, tetanus toxin light chain (TeTx) was expressed in ASER to decrease the synaptic output from the neuron and then the activities of AIA upon salt concentration changes were monitored. We first confirmed that worms expressing TeTx in ASER showed weak but significant defects in taste avoidance learning (Supplementary Fig. 4). After cultivation at high salt in the presence of food, AIA neurons from TeTx-expressing animals responded to salt concentration increase in a manner similar to the wild type, suggesting that the ON response of AIA is regulated by neurons other than ASER (Fig. 4c left, blue line in top panel, middle panel). On the other hand, no significant enhancement was observed in the OFF and ON responses of AIA after food deprivation (Fig. 4c, d). These data suggest that AIA activities in response to salt concentration changes are enhanced after starvation dependent on the output from the ASER neuron.

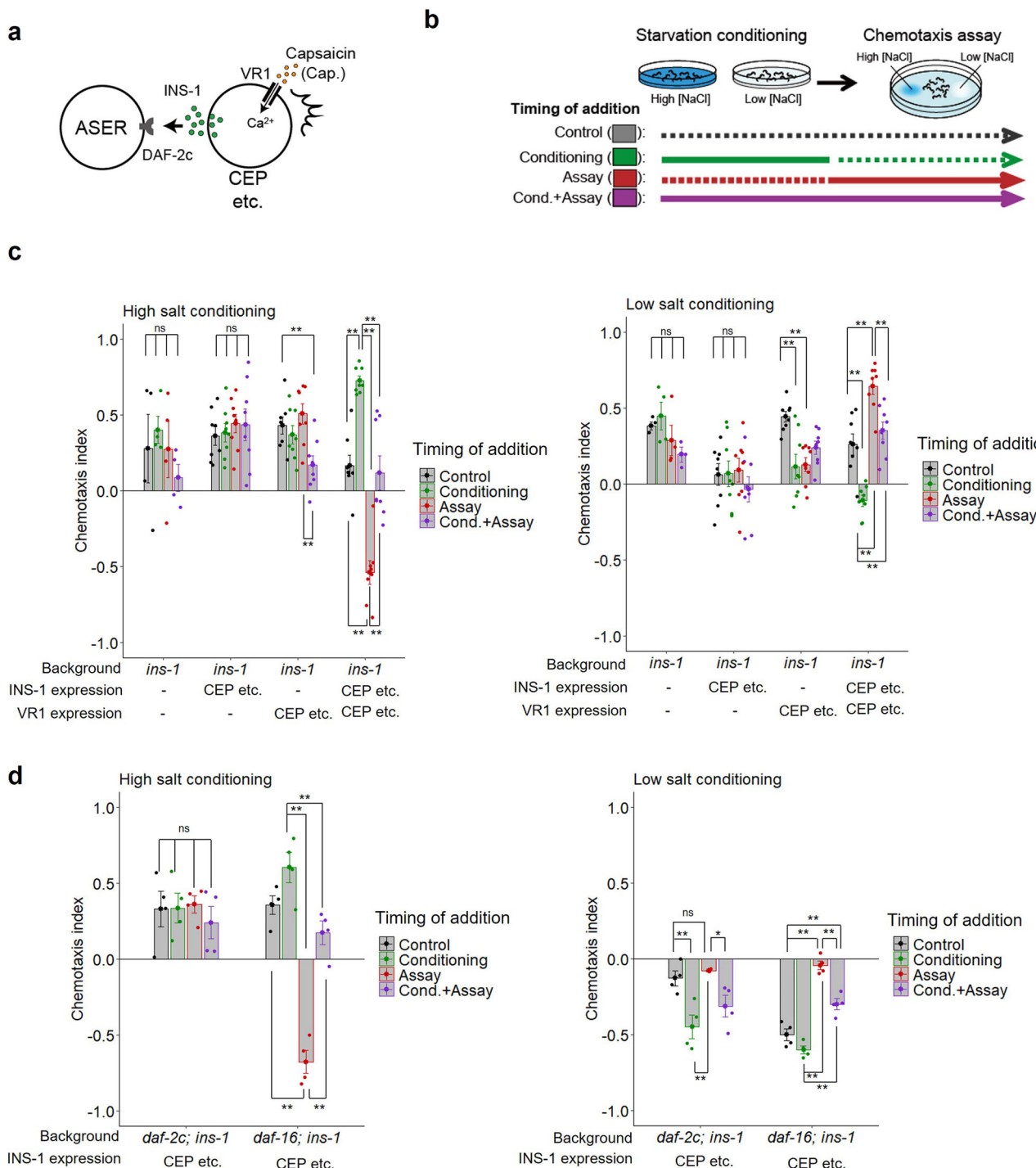

**Fig. 3 INS-1–DAF-2c signaling promotes taste avoidance learning during salt chemotaxis but not during starvation conditioning. a** INS-1::Venus and VR1 were coexpressed in the dopaminergic neurons, such as CEP, and then capsaicin was applied to promote INS-1 release with specific timing. **b** Timing of capsaicin addition during the taste avoidance learning paradigm. **c** Chemotaxis of *ins-1* mutant worms with or without transgenes that drive expression of INS-1::Venus, VR1, or both after high (left) or low (right) salt conditioning in the absence of food. n = 4–8 assays. **d** Chemotaxis of *daf-2c; ins-1* or *daf-16; ins-1* mutant worms expressing both INS-1::Venus and VR1 after high (left) or low (right) salt conditioning in the absence of food. Capsaicin was applied only during salt conditioning (green dots), only during chemotaxis tests (red dots), or during both (purple dots). n = 4 assays. Bars represent mean values; error bars represent SEM. ANOVA with Tukey's post hoc test: *P < 0.05 and **P < 0.01. ns not significant.

**DAF-2c and PLC-1 act in the same genetic pathway to promote high-salt migration after starvation conditioning.** To explore the mechanisms by which the DAF-2c pathway regulates migration toward both higher and lower salt concentrations after starvation conditioning, we conducted epistasis analyses between

*daf-2c* and genes in phosphoinositide metabolism pathways, for which genetic interactions in salt chemotaxis were proposed in our previous studies[26,35]. The PLCβ homolog EGL-8 and the PLCε homolog PLC-1 play major and minor roles, respectively, for regulating the amount of DAG at the ASER axon upon salt

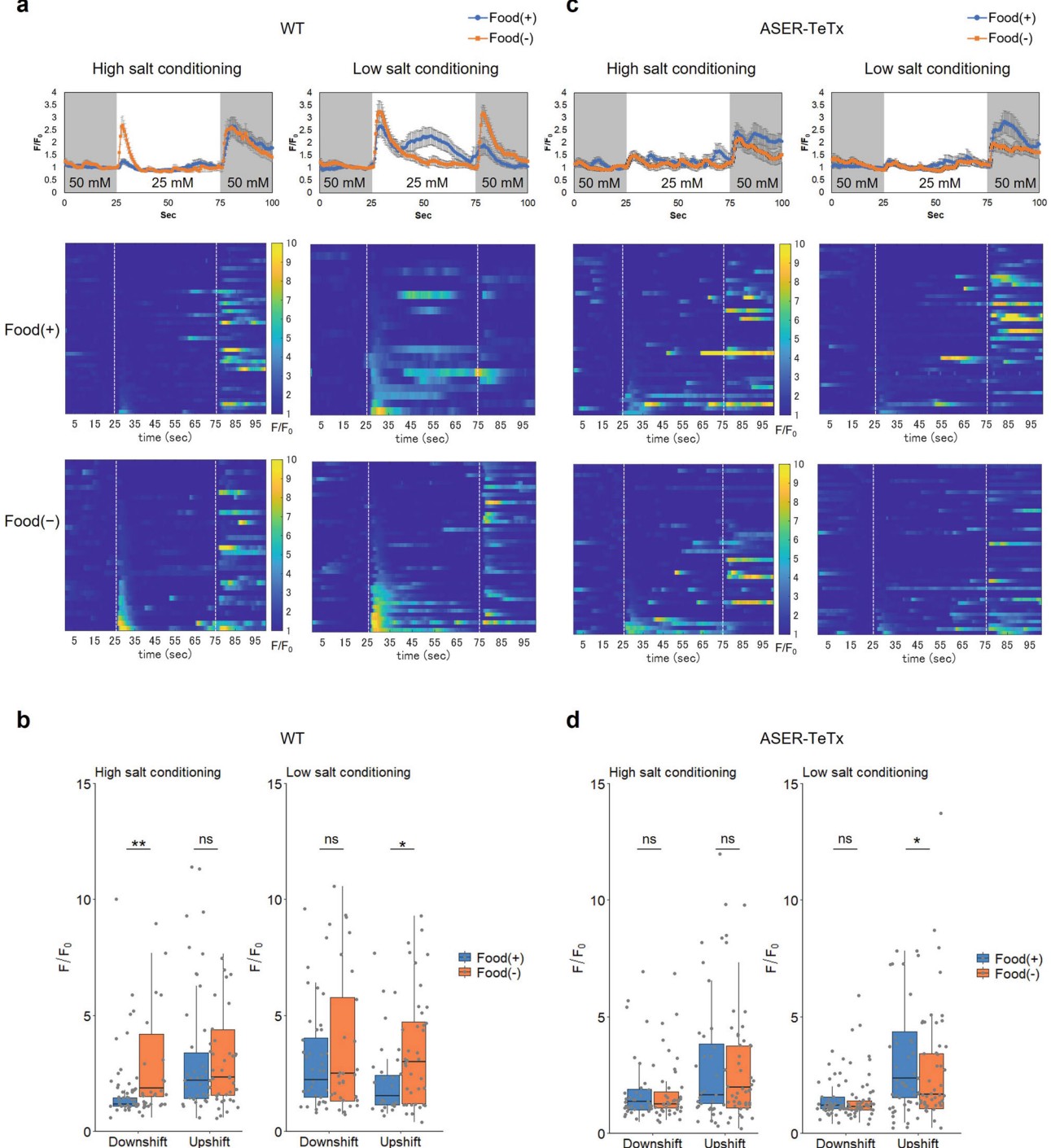

**Fig. 4 Calcium responses of AIA are increased upon salt concentration changes after starvation conditioning.** Calcium responses of AIA upon salt concentration changes after 50 mM (High) or 25 mM (Low) salt conditioning in the presence or absence of food. **a**, **c** Time course of the average fluorescence intensity ratios ($F/F_0$) (top) and heat maps of $F/F_0$ (middle and bottom) of GCaMP6s in AIA. The NaCl concentration was switched from 50 to 25 mM at 25 s and then returned to 50 mM at 75 s. **b**, **d** Quantification of calcium responses of AIA upon a salt concentration decrease (Downshift) or increase (Upshift) after 50 mM (High) or 25 mM (Low) salt conditioning in the presence or absence of food. Peak fluorescence intensity ratios ($F/F_0$) of GCaMP6s in AIA during the 10 s period after the salt concentration change. Calcium responses in the wild type (**a**, **b**: High-salt/Food(+), $n = 43$; High-salt/Food(−), $n = 34$; Low-salt/Food(+), $n = 22$; Low-salt/Food(−), $n = 44$) and in worms expressing tetanus toxin light chain (TeTx) in ASER (**c**, **d**: High-salt/Food(+), $n = 40$; High-salt/Food(−), $n = 40$; Low-salt/Food(+), $n = 44$; Low-salt/Food(−), $n = 46$). Two-tailed Wilcoxon−Mann−Whitney test: *$P < 0.05$ and **$P < 0.01$. ns not significant. Box plots represent the median (central line), the 25th and 75th percentiles (the box), 1.5 times the inter-quartile range from the 25th and 75th percentiles (the whiskers). Individual data points are superimposed on the box plots.

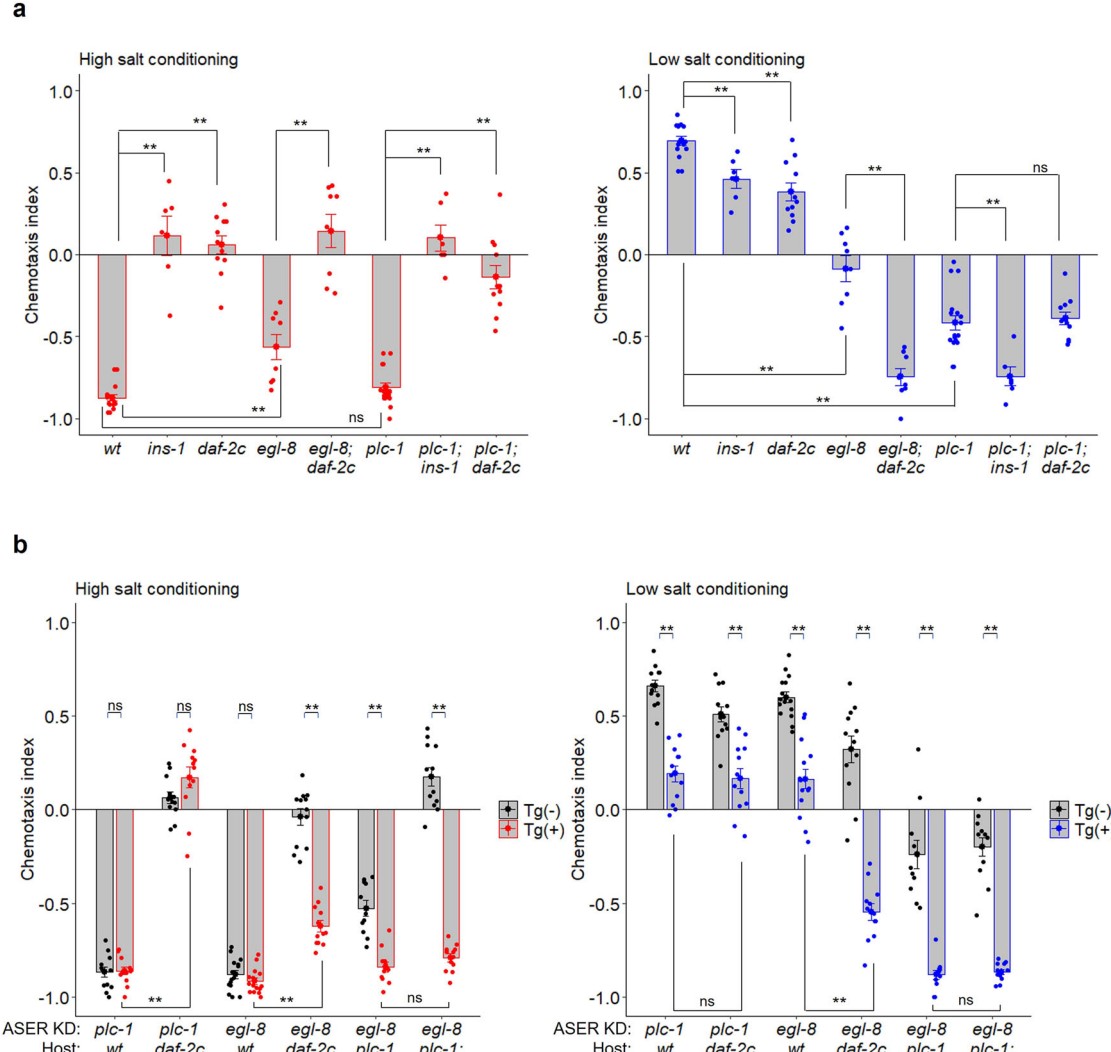

**Fig. 5 The DAF-2c pathway promotes taste avoidance learning via the PLC isozymes PLC-1 and EGL-8. a** Salt chemotaxis after conditioning on agar plates at high or low salt concentrations in the absence of food. Each dot in red or blue represents a chemotaxis index calculated in each chemotaxis assay after conditioning at high or low salt concentrations, respectively. *n* = 6–17 assays. **b** Effects of ASER-specific knockdown of *plc-1* or *egl-8* on salt chemotaxis in the indicated genetic backgrounds. Salt chemotaxis after conditioning on agar plates at high or low salt concentrations in the absence of food using worms expressing Cas9 in ASER and sgRNA with a target sequence of *plc-1* or *egl-8*. Each dot in red or blue represents a value calculated in each chemotaxis assay after conditioning at high or low salt concentrations, respectively. A black dot represents a chemotaxis index calculated in each chemotaxis assay using worms without transgenes. *n* = 11–15 assays. Bars represent mean values; error bars represent SEM. Two-tailed Welch's t-test with Holm correction: **$P < 0.01$. ns not significant.

concentration changes. Starvation conditioning reduces DAG dynamics dependent on the DAF-2 pathway[26]. We tested the effects of the *daf-2c* mutation on the taste avoidance learning of the frameshift deletion mutant *egl-8(n488)* and the *plc-1(pe1238)* mutant, which harbors a deletion in the large isoform of *plc-1*[25]. The *egl-8(n488)* mutant showed reduced taste avoidance after starvation conditioning; the *daf-2c* mutations further enhanced the high-salt migration defect after low-salt conditioning. This suggests that the DAF-2c pathway acts at least partly in parallel with *egl-8* in high-salt migration after starvation (Fig. 5a, right). The *plc-1(pe1238)* mutant exhibited a strong defect in high-salt migration after starvation conditioning at low salt. Interestingly, the *daf-2c* mutation had no additive effect on the high-salt migration defect of *plc-1(pe1238)* (Fig. 5a, right). On the other hand, the mutation of *ins-1*, which acts in both the DAF-2c and DAF-16 pathways (Fig. 1c), further enhanced the high-salt migration defect of *plc-1(pe1238)* after low-salt/starvation

conditioning (Fig. 5a, right). These data suggest that *daf-2c* acts in the same genetic pathway with *plc-1* for driving high-salt migration after starvation conditioning at low salt. On the other hand, the *daf-2c* mutation caused low-salt migration defects after high-salt conditioning both in the *egl-8* and *plc-1* mutant backgrounds (Fig. 5a, left).

**Decreased EGL-8 activity in ASER suppresses aberrant high salt migration in the *daf-2c* mutant after starvation conditioning.** Consistent with the previous report suggesting that EGL-8 functions in several types of neurons, including motor neurons[36], the *egl-8* mutation caused significantly low mobility in salt chemotaxis compared to the wild type (Supplementary Fig. 5). To further clarify the function of PLC-1 and EGL-8 in ASER, we examined knockdown (via the somatic CRISPR/Cas9 method) of the PLC isozymes only in the ASER neuron. ASER-

knockdown of *plc-1* or *egl-8* significantly reduced high-salt migration after starvation conditioning at low salt (Fig. 5b right, 1st and 3rd blocks, Supplementary Fig. 6a, right). ASER-knockdown of *egl-8* further increases low-salt migration bias in the *plc-1(pe1238)* genetic background, suggesting that both PLC isozymes redundantly act in ASER (Fig. 5b, 2nd blocks from right). On the other hand, ASER-knockdown of *egl-8* but not *plc-1* significantly reduced high-salt migration after conditioning with feeding, consistent with the observation that EGL-8 plays a major role in DAG dynamics in the ASER axon after feeding[26] (Supplementary Fig. 6b).

We next examined the effect of ASER-specific knockdown of the PLC isozymes in the *daf-2c(pe2722)* genetic background. ASER-knockdown of *plc-1* significantly reduced high-salt migration after low-salt conditioning in the absence of food in the *daf-2c* mutant, similar to the wild type (Fig. 5b right, 2nd block). Remarkably, the *daf-2c* mutation caused no significant effect on high-salt migration after starvation conditioning with ASER-specific knockdown of *plc-1* (Fig. 5b right, blue bars in 1st and 2nd blocks), which is consistent with the finding that *daf-2c* acts in the same genetic pathway as *plc-1*. ASER-knockdown of *egl-8* suppressed low-salt migration defects and increased migration bias toward low salt in the *daf-2c* mutants after high- and low-salt conditioning, respectively (Fig. 5b, 4th blocks). Furthermore, the *daf-2c* mutation caused no significant effect on low-salt migration after starvation conditioning at high salt with knockdown of *egl-8* and the *plc-1* mutation (Fig. 5b, left, red bars in 1st and 2nd blocks from right). These data imply that DAF-2c promotes migration toward lower salt via suppression of both EGL-8 and PLC-1 in ASER (Fig. 7).

### Requirement of putative Akt phosphorylation sites of PLC-1(L) for high-salt migration after starvation conditioning.

PLC-1 exists as multiple isoforms (WormBase; https://wormbase.org) that can be classified into short isoforms, PLC-1(S), and long isoforms, PLC-1(L), which harbor short and long N-terminal tails, respectively[25] (Fig. 6a). PLC-1(L) is required for taste avoidance learning: the *plc-1(pe1238)* mutant, which carries a deletion in the region specific for PLC-1(L)[25], showed strong defects in high-salt migration after starvation conditioning (Fig. 5a). We noticed three Akt phosphorylation consensus sequences, RXRXXT, in the long N-terminal tails of PLC-1(L) (Fig. 6a). We next examined the effects of mutations in Akt homologs on the taste avoidance learning of *daf-2c*. A deletion mutation of *akt-1*, namely *akt-1(ok525)*, resulted in a strong defect in taste avoidance learning (Fig. 6b). In contrast, a deletion mutation of *akt-2*, namely *akt-2(ok393)*, resulted in normal taste avoidance learning. The *daf-2c* mutation did not further enhance the *akt-1* mutant and the *akt-1* mutation was epistatic to the *daf-2c* mutation in taste avoidance learning. These results are consistent with the notion that DAF-2c regulates taste avoidance learning through Akt (Fig. 6b). We further examined the putative Akt phosphorylation threonine residues by expressing either the wild-type PLC-1(L) or mutant form of PLC-1(L), namely PLC-1(3A) in which all of the threonine residues are mutated to alanine, in the *plc-1(pe1238)* mutant and then assessing the effects on taste avoidance learning. Wild-type PLC-1 expression significantly mitigated the defect in high-salt migration after low-salt/starvation conditioning; however, PLC-1(3A) expression had no significant effect on low-salt/starvation conditioning (Fig. 6c, right). These results imply that the DAF-2c/ATK-1 pathway phosphorylates PLC-1(L) under starvation conditioning, thereby promoting migration toward higher salt to avoid salt concentrations associated with starvation (Fig. 7).

## Discussion

Our cell-specific rescue or knockdown experiments suggest that INS-1 secretion from AWA and ASI chemosensory neurons and AIA interneurons promotes taste avoidance learning. These neurons express endogenous INS-1, with strong expression shown in AIA (CeNGEN; https://cengen.shinyapps.io/CengenApp/); they connect to each other by gap junctions and send synaptic outputs to ASER[37] (Supplementary Fig. 2d). As INS-1 is localized to presynaptic regions[9], our findings suggest that local release of INS-1 from multiple neuron types may immediately modulate the functions of ASER through the axonal DAF-2c receptor. The axonal transport of DAF-2c is increased during starvation[10]. This localization change of DAF-2c may be required for acquisition phase of learning. In the present study, activation of INS-1–DAF-2c signaling during chemotaxis led to taste avoidance after starvation conditioning. Therefore, increased DAF-2c signaling at the axon during starvation and its persistent activation during chemotaxis might be required to induce and maintain starvation-induced taste avoidance.

Strong activation of AIA neurons was observed upon a salt concentration change, i.e., increased or decreased salt concentrations after high- or low-salt conditioning in the presence of food, respectively. Given previous evidence that artificial activation of AIA suppresses turning behavior[38], it is possible that turning is suppressed by AIA activation upon salt concentration changes to those that the worms experienced during feeding and that this contributes to the observed attraction toward given salt concentrations. In contrast, strong AIA responses were observed upon concentration changes to both low and high salt after starvation conditioning; these responses were dependent on ASER. It was reported that the spontaneous activities of AIA were increased under starvation conditions dependent on inputs from chemosensory neurons[33]. During chemotaxis where animals experience multiple rounds of increase and decrease of salt concentrations, this effect of starvation is likely augmented. These findings imply that increased AIA activities may increase INS-1 secretion during chemotaxis after starvation, which plays a key role in maintaining the high activity of DAF-2c signaling by sending feedback signals to DAF-2c in ASER. We confirmed that worms expressing TeTx in ASER showed a weak but significant defect in high-salt migration after low-salt conditioning in the absence of food (Supplementary Fig. 4). Synaptic output from ASER and possible feedback regulation of INS-1 from AIA may be more important for chemotaxis after low-salt conditioning than that after high-salt conditioning. Consistent of this, we have previously shown that stronger synaptic transmission was observed in ASER after low salt exposure in buffer than that after high salt exposure[34].

In addition to the action of INS-1 on DAF-2c, INS-1 acts on the DAF-2a–DAF-16 pathway in high-salt migration after starvation conditioning at low salt. Similar to the action of INS-1 on DAF-2–DAF-16 signaling in the intestine and AWC chemosensory neurons[20,39], INS-1 may act as an antagonist on DAF-2a in ASER for the regulation of taste avoidance learning. Conversely, mutant phenotypes and epistasis analyses suggest that INS-1 acts on DAF-2c as an agonist. Taken together, these findings imply that INS-1 acts on both DAF-2c and DAF-2a as an agonist and antagonist, respectively, to effectively regulate behavioral plasticity at different cellular sites in ASER. We previously reported that a shift to the restrictive temperature of the temperature-sensitive *daf-2(e1370)* mutant, in which functions of both DAF-2a (E11.5−) and DAF-2c (E11.5+) isoforms are disrupted, during either starvation conditioning or chemotaxis assay caused defects in starvation-induced avoidance of odorant[13]. Therefore, *daf-2* is required for conditioning, as well as chemotaxis. In the ASER neuron, DAF-2a (E11.5−) represses starvation-dependent nuclear

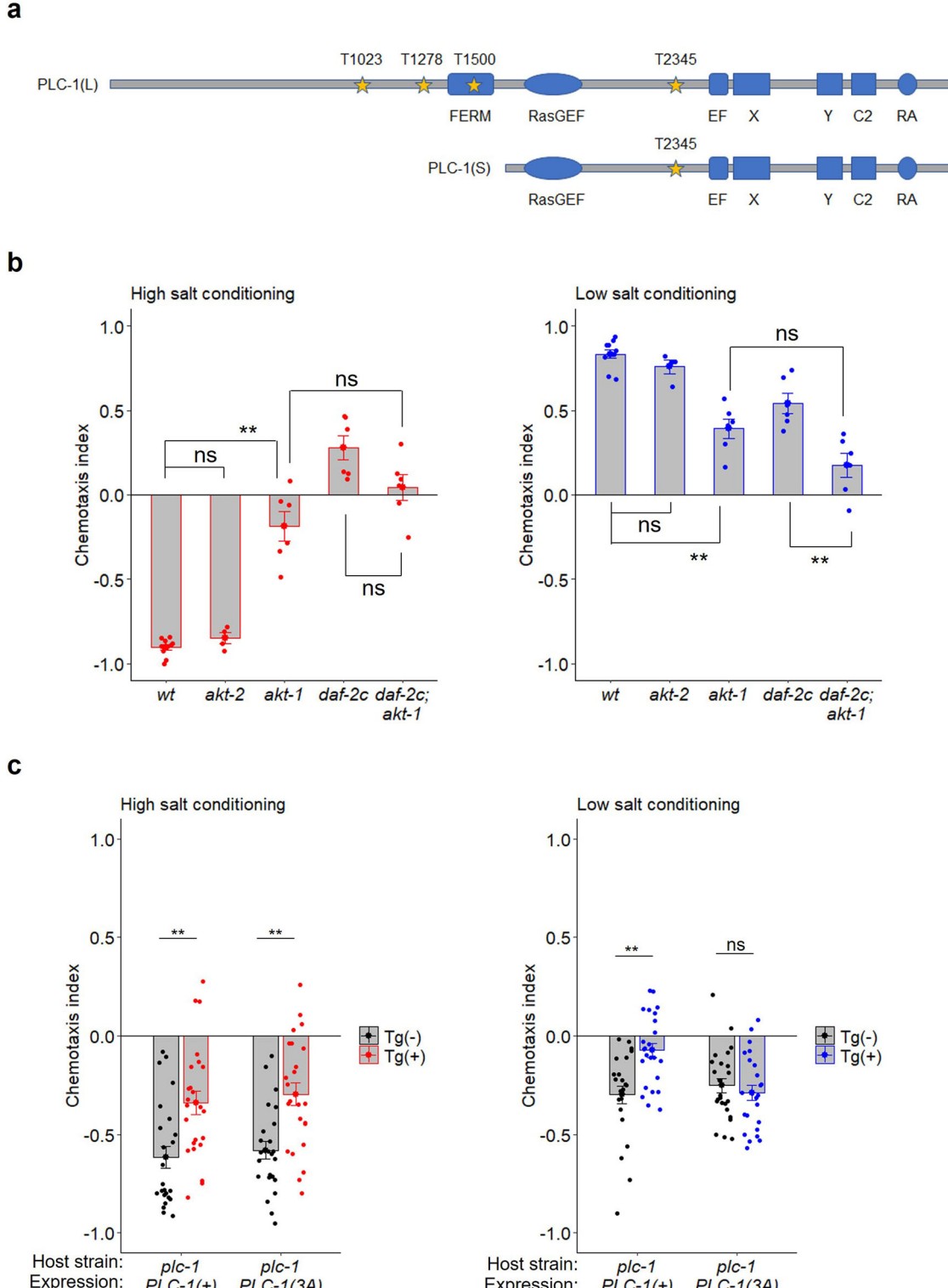

**Fig. 6 Putative Akt phosphorylation sites of PLC-1(L) are required for high-salt migration after starvation conditioning at low salt. a** The domain structures of PLC-1(L) and PLC-1(S) were analyzed with Pfam 33.1 (http://pfam.xfam.org/). Threonine residues in the Akt phosphorylation consensus sites (RXRXXT) are highlighted with yellow stars. Genetic interactions between Akt and *daf-2c* mutations (**b**, n = 4–10 assays) and rescue of taste avoidance learning by PLC-1(+) or PLC-1(3A) in the *plc-1(pe1238)* mutant (**c**, n = 24 assays). Chemotaxis indices after high- or low-salt conditioning in the absence of food. Each dot in red or blue represents a value calculated in each chemotaxis assay after conditioning at high or low salt concentrations, respectively. A black dot represents a chemotaxis index calculated in each chemotaxis assay using worms without transgenes. Bars represent mean values; error bars represent SEM. Two-tailed Welch's t-test with Holm correction: **P < 0.01. ns not significant.

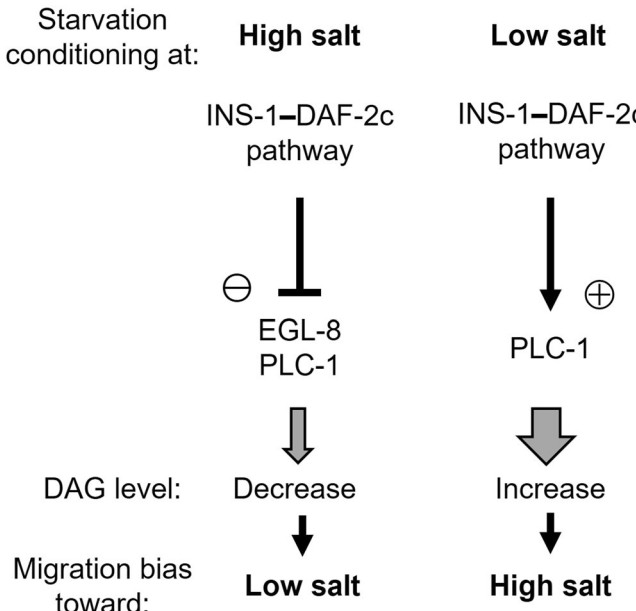

**Fig. 7 The INS-1–DAF-2c pathway promotes learned taste avoidance via distinct PLC isozymes.** Worms migrate toward both higher and lower salt concentrations to effectively avoid the salt concentrations encountered under starvation conditions. We propose a model in which the INS-1–DAF-2c pathway exerts distinct effects on the PLC isozymes EGL-8 (PLCβ) and PLC-1 (PLCε) after starvation conditioning. We postulate that the INS-1–DAF-2c pathway suppresses EGL-8 and PLC-1 activities and elevate PLC-1 activity, which decreases and increases DAG levels, and thereby promotes both lower and higher salt migrations, respectively.

localization of DAF-16, implying that the DAF-2a–DAF-16 pathway functions in starvation conditioning[12]. Taken together with the finding that excess INS-1 release only during conditioning reduced learned taste avoidance (Fig. 3a–c), INS-1 secretion of the appropriate amount at the appropriate timing is required for the action of the DAF-2a–DAF-16 pathway in starvation conditioning.

In our previous study, we showed that starvation reduced DAG dynamics at the ASER axon upon salt concentration changes, thereby reducing the characteristic DAG reponses that drive the animal to the previously experienced salt concentration[26]. This starvation-dependent change in DAG dynamics was defective in mutants of *daf-2*. EGL-8 (a PLCβ) plays an essential role in DAG dynamics[26]; we found that ASER-specific knockdown of *egl-8* promoted low-salt migration and suppressed aberrant high-salt migration in *daf-2c* mutants. Thus, DAF-2c signaling seems to reduce EGL-8-dependent DAG production, whereby increasing the migration bias toward lower salt after starvation conditioning at high salt. Furthermore, another genetic pathway was demonstrated in the present study in which DAF-2c–PLC-1 signaling increases migration bias toward higher salt after starvation conditioning at low salt. Given that mutations of the putative Akt phosphorylation sites diminished the action of PLC-1 in high-salt migration after starvation conditioning, DAF-2c may increase the activity of PLC-1 (a PLCε) by its phosphorylation via AKT-1. Although PLC-1 plays a minor role in DAG dynamics at the ASER axon upon salt concentration changes after feeding, a high-salt migration defect caused by ASER-specific knockdown of *plc-1* was comparable to that of *egl-8* after starvation conditioning; thus, PLC-1 seems to play an essential role after starvation. The ASER-specific knockdown of *plc-1* caused stronger defects in high-salt migration than were caused by the *daf-2c* mutation, indicating that basal PLC-1 activity and/or PLC-1 activation by

other unknown signaling pathways further increases migration bias toward higher salt after starvation. PLCβ is known to be directly activated by Gαq[40]. Also, in *C. elegans*, EGL-30 (Gαq) is proposed to activate EGL-8 in several types of neurons, including ASER[25]. On the other hand, PLCε is activated by diverse regulators, such as small G proteins[40]. Although regulators for PLC-1 activity are mostly unclear, multiple inputs, including DAF-2c/Akt signaling, may regulate PLC-1 to control taste avoidance after starvation.

In addition to the difference in the activators between PLCβ and PLCε, mechanisms for membrane binding and cellular distribution are different. The C-terminal domain (CTD) characteristic for PLCβ is known to determine membrane binding and spatial localization to regions of the membrane enriched in the substrate, phosphatidylinositol-4,5-bisphosphate ($PIP_2$)[40]. Unlike PLCβ, the membrane-binding regions of PLCε are obscure, although the pleckstrin-homology (PH) domain is proposed to bind $PIP_2$ and anchor the enzyme at the membrane[40]. In this study, we found the FERM domain in the N-terminal region specific for long isoform of PLC-1 (Fig. 6a). The FERM domain is involved in localization to the plasma membrane and the purified FERM domain binds to $PIP_2$[41]. As a putative Akt-phosphorylation site exists in the FERM domain of PLC-1, DAF-2c/Akt signaling may regulate membrane localization of PLC-1 via phosphorylation of this site, which needs to be confirmed in future studies. The possible differences in upstream regulators and localization patterns in the plasma membrane might contribute to the separate roles of EGL-8 and PLC-1 in DAF-2c-dependent salt chemotaxis plasticity. The ASER neuron has synaptic connections to various interneurons and expresses a variety of neuropeptides, which could extrasynaptically modulate neural circuits. In ASER, each of PLCs may modulate ASER functions, such as synaptic transmission, at the different sites in the axon along with DAF-2c. It will be interesting to elucidate how multiple PLCs modulate ASER functions and its downstream neural circuits to regulate dramatic changes in salt chemotaxis after starvation.

## Methods

**Caenorhabditis elegans strains**. *Caenorhabditis elegans* Bristol strain N2 was used as the wild type. The *C. elegans* strains were grown and maintained on NGM plates seeded with *Escherichia coli* strain NA22 at 20 °C, except for in calcium imaging experiments in which *E. coli* strain OP50 was used as a food source. Standard genetics methods were used to generate strains with multiple mutations by crossing. The mutants and transgenic worms used in this study are listed in Supplementary Data 1.

**DNA constructs and transgenesis**. To generate INS-1::Venus expression plasmids, we generated the destination vector carrying *ins-1::venus* cDNA, namely pDEST-INS-1::Venus, and then inserted promoter sequences upstream of *ins-1::venus* via the LR reaction with entry vectors carrying the promoter sequences (pENTR-promoter) using the GATEWAY cloning technique (ThermoFisher). We used the *daf-7* promoter (3.1 kb) for ASI expression, the *odr-10* promoter (1.1 kb) for AWA expression, the *gcy-28d* promoter (2.8 kb) (a gift from Dr. Takeshi Ishihara) for strong AIA and weak ASI expression, the *trx-1* promoter (1.0 kb) for ASJ expression, the *glr-3* promoter (2.1 kb) for RIA expression, the *tph-1* promoter (1.7 kb) for ADF and NSM expression, the *tbh-1* promoter (4.5 kb) for RIC expression, and the *dat-1* promoter (0.7 kb) for CEP, ADE, and PDE expression. The expression patterns were confirmed by INS-1::Venus fluorescence using confocal microscopy.

We generated an *ins-1* fosmid reporter according to the protocol reported by Tursun et al.[42]. The BALU9 cassette (a gift from Dr. Oliver Hobert), which contains *sl2::gfp* cDNA and the FgF (FRT-galK-FRT) sequence, was inserted downstream of the *ins-1* ORF in a fosmid vector (WRM0618bG05).

To generate a plasmid containing rat *VR1* cDNA (a gift from Dr. Cori Bargmann) driven by the *dat-1* promoter, we generated the destination vector carrying rat *VR1* cDNA, namely pDEST-VR1, and then inserted the *dat-1* promoter sequence upstream of *VR1* via the LR reaction with the entry vector carrying the *dat-1* promoter (pENTR-dat-1) using the GATEWAY cloning technique (ThermoFisher).

To generate plasmids containing Cas9 cDNA (a gift from Dr. Bob Goldstein) driven by the *rgef-1* promoter (3.5 kb) (a gift from Dr. Roger Pocock), *gcy-28d* promoter (2.8 kb), *ges-1* promoter (3.2 kb), or *gcy-5* promoter (2.0 kb), we

generated the destination vector carrying Cas9 fused to *SL2::CFP*, namely pDEST-Cas9::SL2::CFP, and then inserted the promoter sequences upstream of Cas9 via the LR reaction with the entry vectors carrying the promoters, i.e., pENTR-rgef-1, pENTR-gcy-28d, pENTR-ges-1, and pENTR-gcy-5, respectively. We used two rounds of inverse PCR to generate plasmids for expression of single-guide RNA (sgRNA) driven by the *U6* promoter. First, we generated a plasmid for expression of sgRNA without a target sequence under the control of the *U6* promoter, namely *U6p::sgRNA(empty)*, using the pDD162 plasmid (a gift from Dr. Bob Goldstein) as a template. Second, we added the following target sequences to *U6p::sgRNA(empty)*: *ins-1*, caccaccaacaaaagcgagg; *plc-1* #1, tcgtcgagaattggagtggg; *plc-1* #2, ggaagaacaccaggaactgg; *egl-8* #1, atatgggaagctcgcacggg; and *egl-8* #2, ctaaatgacttgggtcttgg. We confirmed that the restricted expressions of Cas9::SL2::CFP were driven in the nervous system, AIA and the intestine, the intestine, chemosensory neurons, including ASH, ADL and ASI, RIC or ASER by the *rgef-1* promoter, *gcy-28d* promoter, *ges-1* promoter, *gpc-1* promoter, *tbh-1* promoter or *gcy-5* promoter, respectively, based on CFP fluorescence imaging achieved using confocal microscopy (Supplementary Fig. 3). The effects of knockdown in somatic cells were confirmed by pan-neuronal Cas9 expression using the *rgef-1* promoter with sgRNA for each target; confirmation of knockdown was followed by the taste avoidance learning assay (Supplementary Fig. 7).

To generate a plasmid for expression of GCaMP6s or mCherry in AIA, the *ins-1(short)* promoter was inserted upstream of GCaMP6s or mCherry cDNA via the LR reaction with the destination vector carrying GCaMP6s, i.e., pDEST-GCaMP6s (a gift from Dr. Takeshi Ishihara), or pDEST-mCherry, respectively, and the entry vector carrying the *ins-1(short)* promoter, i.e., pENTR-ins-1. To generate a plasmid for the expression of TeTx, the *gcy-5* promoter was inserted upstream of TeTx cDNA via the LR reaction with the destination vector carrying TeTx, namely pDEST-TeTx, and pENTR-gcy-5.

To generate a plasmid for expression of a long PLC-1 isoform, i.e., PLC-1(L), which resembles the *plc-1i* isoform (https://wormbase.org/) but with extension of exon 9, in ASER, the *gcy-5* promoter was inserted upstream of PLC-1(L) via the LR reaction with the destination vector carrying PLC-1(L), namely pDEST-PLC-1(L)[25], and pENTR-gcy-5. To generate a plasmid for expression of a mutant form of PLC-1(L), namely PLC-1(3 A), in ASER, we first mutated three putative Akt phosphorylation threonine residues (T1023, T1278, and T1500) to alanine residues by three rounds of inverse PCR using PLC-(L) as a template; we then generated the destination vector pDEST-PLC-1(3A) and inserted the *gcy-5* promoter upstream of PLC-1(3A) via the LR reaction.

DNA constructs except for the *ins-1* fosmid reporter were injected at concentrations of 5–90 ng/μl along with a coinjection marker, i.e., *lin-44p::mCherry* (20 ng/μl), *myo-3p::GFP* (30 ng/μl) or *myo-3p::Venus* (10 ng/μl), and a carrier DNA, namely pPD49.26. Injection mixtures were prepared to a final concentration of 100 ng/μl. The *ins-1* fosmid reporter was injected at 15 ng/μl along with a co-injection marker, pRF4 (2 ng/μl), and the *C. elegans* genome (150 ng/μl) as a carrier DNA.

**Salt concentration learning assay.** Salt concentration learning assays were performed according to our previously reported procedures[9,11]. For conditioning on agar plates, adult worms were transferred to NGM plates with 25- or 100 mM NaCl in the absence or presence of a bacterial food (NA22) for 4–5 h. After conditioning, the worms were placed at the center of a 9 cm test plate (25 mM potassium phosphate at pH 6.0, 1 mM CaCl$_2$, 1 mM MgSO$_4$, and 2% agar) with an NaCl gradient from 35 to 95 mM and allowed to crawl for 45 min[11]. For conditioning in buffer solutions, adult worms were transferred to a conditioning buffer (5 mM potassium phosphate at pH 6.0, 1 mM CaCl$_2$, and 1 mM MgSO$_4$) with or without 20 mM NaCl. After conditioning, the worms were placed at the center of a 9 cm test plate (5 mM potassium phosphate at pH 6.0, 1 mM CaCl$_2$, 1 mM MgSO$_4$, and 2% agar) with an NaCl gradient which had been formed overnight by placing an agar plug (5 mm diameter) containing 100 mM NaCl close to the edge of the plate and allowed to crawl for 30 min[9]. Fifteen to two hundred worms were used in each assay. Chemotaxis index and immobility index were determined according to the following equation:

$$Chemotaxis\,index = (N_A - N_B)/(N_{all} - N_C),$$

$$Immobility\,index = N_C/N_{all}$$

where $N_A$ and $N_B$ are the number of worms in the high- and low-salt areas, respectively, $N_{all}$ is the total number of worms on a test plate, and $N_C$ is the number of worms in the area around the starting position. A learning index was determined by subtracting the chemotaxis index determined after conditioning in a buffer with 20 mM NaCl from that determined without NaCl in the absence of food (Supplementary Fig. 2a).

For the salt concentration learning assay shown in Figs. 2 and 3, 100 μM of capsaicin or solvent (70% EtOH as a control) was added to the NGM plates during conditioning and/or to the test plates during chemotaxis assays.

**Calcium imaging.** Calcium imaging was conducted according to our previously reported procedure[43]. We expressed a calcium indicator, namely GCaMP6s, along with mCherry in AIA neurons using the *ins-1(short)* promoter. For conditioning, adult worms were transferred to NGM plates with 25- or 50 mM NaCl in the absence or presence of a bacterial food (OP50) for 6 h. Worms were then physically immobilized in a microfluidic device and an imaging buffer (25 mM potassium phosphate at pH 6.0, 1 mM CaCl$_2$, 1 mM MgSO$_4$, and 0.02% gelatin; osmolarity was adjusted to 350 mOsm with glycerol) containing 50 mM NaCl was delivered to the tip of the nose. The NaCl concentration contained in the imaging buffer was shifted from 50 to 25 mM and then recovered to 50 mM after 50 s. Fluorescence intensities of GCaMP6s and mCherry were simultaneously monitored at a rate of two frames per second. Fluorescence intensities in the neuronal processes of AIA were analyzed with custom-made scripts using ImageJ software. The average fluorescence intensity of the ratio of GCaMP6s to mCherry over 20 frames (10 s) prior to the salt concentration shift from 50 to 25 mM was set as $F_0$, and the fluorescence intensity ratio of GCaMP6s to mCherry relative to $F_0$ ($F/F_0$) was calculated for a time series of images.

**Statistics and reproducibility.** Statistical analyses were done using R3.6.1 (http://www.R-project.org/). For multiple comparison, Welch's t-test with Holm correction or ANOVA followed by Dunnett's or Tukey's post hoc test was applied. All analyses were performed with at least four biological replicates. In behavioral assays using transgenic *C. elegans* strains, a minimum of two independent transgene arrays were tested for effects on behaviors, and then all of the obtained data were used for quantification. Non-parametric Wilcoxon−Mann−Whitney test was applied to compare calcium responses of AIA neurons.

**Reporting summary.** Further information on research design is available in the Nature Research Reporting Summary linked to this article.

## Data availability

Source data for graphs are available in the figshare repository (https://doi.org/10.6084/m9.figshare.16791640) (ref. [44]). All other data are available from the corresponding author on reasonable request.

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

## Acknowledgements
We thank Dr. Cori Bargmann for rat *VR1* cDNA, Dr. Bob Goldstein for the pDD162 plasmid, Dr. Takeshi Ishihara for GCaMP6s cDNA and the *gcy-28d* promoter, Dr. Oliver Hobert for the pBALU9 cassette, Dr. Roger Pocock for the *rgef-1* promoter, and the *Caenorhabditis* Genetics Center and the National Bioresource Project (Japan) for strains. We are grateful to Dr. Ryo Iwata and Dr. Yu Toyoshima for sharing unpublished results regarding *plc-1* and custom scripts for MATLAB, respectively. This work was supported by JSPS KAKENHI (Grant Numbers: JP19H04947, JP17H06113, JP25870172, and JP21700345) and UTokyo Center for Integrative Science of Human Behavior (CiSHuB).

## Author contributions
M.T. conceived the project; M.T. and M.S.J. designed and conducted the experiments; all authors interpreted the results; M.T. wrote the manuscript with input from Y.I.

## Competing interests
The authors declare no competing interests.
