## [Transparent Peer Review File · Communications Biology]

Reviewers' comments:

Reviewer #1 (Remarks to the Author):

In this manuscript, Tomioka et al study the function of a *daf-2c* signaling pathway in salt avoidance after conditioning with starvation. The study was built on their earlier findings showing that an isoform of *daf-2*, *daf-2c*, acts in the process of the salt-sensing neuron ASER to regulate salt avoidance learning and that an insulin peptide INS-1 released from several neurons including interneurons AIA and RIC regulates salt learning. In this manuscript, the authors further show that INS-1 released during the salt chemotaxis assay after conditioning is sufficient for animals to show salt avoidance. They also show that conditioning increases AIA calcium response to the salt concentration changes consistent with the behavioral changes in salt chemotaxis. Furthermore, they identify separate functions of PLC beta (*egl-8*) and PLC epsilon (*plc-1*) in salt learning and both of them are required to function in ASER downstream of *daf-2c* via Akt. In order to dissect the roles of these signaling pathways in salt learning, the authors used multiple approaches, including cell-specific knock out/down and temporal manipulation of dopamine signals. The genetic analysis is also thorough. The findings are clearly presented and they support the authors' conclusions. The following suggestions would help to increase the clarity of the manuscript.

Line 70 – 73, "The responses of AIA interneurons, which are downstream of ASER in the neural circuit and a main site of INS-1 release, to changes in salt concentration are elevated after starvation conditioning dependent on the synaptic output of ASER; thus, the feedback signal of INS-1 from AIA to DAF-2c in ASER during salt chemotaxis seems to strengthen DAF-2c signaling." The changes in AIA activity are supported by their results. However, it is not clear whether the results suggest the strengthening of the *daf-2c* signaling as the authors state here. It will be helpful to further clarify this point.

In Fig1b, it is helpful to show the comparison between *daf-2c* and wt, and between *daf-16* and wt.

In Fig2c, the authors manipulate dopamine signals to interrogate the timing for INS-1 function. A couple of results for high salt conditioning need more clarification. In group 4, the 4th column is significantly different from the 3rd column and in group 5 the 4th column is different from the 1st (control). Similarly, in Fig3c for high salt conditioning, the 4th column in group 3 shows a significant effect. Some interpretation of these results during the description of the experiments will be very helpful for the clarity of the paper.

Do Fig4 and Supplemental Fig4 show the results from the same experiments? If so, it will be much easier to read if the results in Supplemental Fig4 are included in Fig4.

Fig5a, it will be more clear to show the comparison between *elg-8* and wt, and also between *plc-1* with wt.

It is interesting that *elg-8* and *plc-1* have different functions in salt learning, although they both function downstream of *daf-2c* and akt. It will be informative if the authors discuss and speculate possible mechanisms for their separate roles.

Reviewer #2 (Remarks to the Author):

C. elegans worms learn to prefer past experienced concentrations of salt if reared in the presence of food. However, worms switch their preference towards low/high concentrations if conditioned in the absence of food. Here, the authors made an impressive effort to delineate the molecular details that underlie learning within individual neurons of *C. elegans* worms. Specifically, they showed that INS-1 secretion during animal behavior is promoting the dispersal from the past experienced salt concentrations. This is probably mediated by the *daf-2c* receptor expressed on the ASER neuron, which is also the main sensor for the salt. Downstream to DAF-2c, Akt signaling promotes dispersal via PLC-beta or PLC-epsilon, depending on the concentration the worms were raised in.

These mechanistic molecular and cellular details are of importance to the community and for the learning and memory field in general. The experimental results and the conclusions drawn from them sound valid and so are the statistical analyses. The methods are sufficiently detailed to allow successful reproduction of the results.

There are a couple of issues that the authors could address:

1. The authors claim that INS-1 secretion drives activity of DAF-2c during the chemotaxis assay to avoid the learned salt concentration. However, according to figure 3d, *daf-2c* mutants could not be rescued to show the wt phenotype. So, another possibility is therefore that *daf-2c* mutants did not learn the experience from the first place. Is there another piece of evidence that can reconcile this point? Perhaps the authors could acknowledge and discuss this possibility in the discussion section.

2. Similarly, it is claimed that AIA activities are enhanced after starvation and depend on the output from the ASER neuron (lines 213-214). However, synaptic outputs from ASER during learning were inhibited (TeTx). So maybe the animals failed to learn or perhaps the learning process was not effective enough. In this case, the authors could control this by assaying the behavior of ASER TeTx-expressing animals (or chemogenetically inhibit ASER during the learning period only).

Minor:

1. Note some of the refs are clipped (e.g., #22 #39).
2. Line 84: should read "animals with a large deletion...."
3. Line 99 and line 104: there is a reference to panel d in figure 1, but there is no panel d.
4. In legend to figure 1, it will be helpful to add how CI was calculated and what does negative and positive CI mean (it appears in the methods but will be good to mention here as it repeatedly appears throughout the MS).
5. Figure 2b, will be helpful to add to the legend that continuous line denotes with capsaicin and dotted line without capsaicin.

Reviewer #3 (Remarks to the Author):

Previous work from Iion's lab has revealed the importance of insulin/PI3K signaling in salt chemotaxis learning, and discovered one particular splicing isoform of insulin receptor, *daf-2c*, localizing to the axonal processes of salt sensing neuron ASER to regulate salt aversive learning. In the present study, the authors carry out a detailed genetic analysis of DAF-2c signaling in salt chemotaxis learning. They dissect the site of action of INS-1, and show that INS-1/DAF-2c signaling acts during salt chemotaxis to modulate high- or low- salt migration after starvation conditioning. Calcium imaging analysis demonstrates that synaptic output of ASER neuron is indispensable for the responses of AIA interneurons to the changes of salt levels after starvation. They further dissect how different phospholipase C participate in salt learning in the context of *daf-2c* signaling, and identify the potential Akt phosphorylation sites on PLC, which are critical for taste aversive learning.

The findings in this study is interesting, and the results are convincing. These observations are likely important for the understanding of INS-1/DAF-2c signaling in general. It further strengths their earlier observations and conclusions. The experiments are strictly controlled. The manuscript will potentially attract broader audiences including those that study insulin signaling, neural plasticity and behavior. Overall, this paper addresses an important question with solid genetic data.

I have a few concerns with the presentation of the data.

- 1). Fig. 1d does not seem to exist. Lines 99, 104 and 244 refer to 1d, which is not displayed.
- 2). For the comparison purpose, it is worth including *daf-16* in the high salt conditioning model in the fig 1c, which acts in parallel to *daf-2c*.
- 3). Capital A in fig 1 legend is a typo.
- 4). Even though it was suggested that AIA is a main site of *INS-1* release, the learning defect of *ins-1* was not fully rescued by expressing *ins-1* in AIA neurons. Instead, *ins-1* expression in AWA, ASI or RIC gives a much better rescue (Supplementary fig 2). In supplementary fig 3, it also shows that cell-specific knockdown of AIA does not lead to a dramatic change in taste avoidance learning. This might be caused by inefficient knock-down of *ins-1* in AIA by crispr, but it could reflect the contribution for the other neurons. Authors are encouraged to knock-down *ins-1* in AWA, ASI and RIC even though its expression in RIC has not been confirmed. The *dat-1* single mutant was not investigated in the article, but it has been reported to be defective in salt learning. This might be an indirect evidence that RIC neurons may be involved.
- 5). Please include the number of animals (n=?) imaged in the fig 4 legend.
- 6). *egl-8* mutants behave differently in low and high salt migration after high and low salt conditioning (fig 5a). In low salt migration after high salt conditioning, *egl-8*; *daf-2* double is similar to *daf-2* single. However, in high salt migration after low salt conditioning, the double is significantly different from either single. It might not be appropriate to draw one conclusion for two different observations (lines 238 to 241).

Point-by-point responses to the referees' comments

Response to Reviewer #1 :

We are grateful for the reviewer's comments. It was very helpful to improve our paper.

Line 70 – 73, “The responses of AIA interneurons, which are downstream of ASER in the neural circuit and a main site of INS-1 release, to changes in salt concentration are elevated after starvation conditioning dependent on the synaptic output of ASER; thus, the feedback signal of INS-1 from AIA to DAF-2c in ASER during salt chemotaxis seems to strengthen DAF-2c signaling.” The changes in AIA activity are supported by their results. However, it is not clear whether the results suggest the strengthening of the daf-2c signaling as the authors state here. It will be helpful to further clarify this point.

As the reviewer pointed out, we do not have direct evidence showing that DAF-2c signaling is strengthened by AIA activity following ASER activation upon salt concentration changes. In this paper, we would like to propose a possible feedback regulation of INS-1 – DAF-2c signaling for learned behavior. Based on the following findings, we speculate that increased activation of AIA during chemotaxis stimulates INS-1 release from AIA, and INS-1 presumably acts on DAF-2c in ASER, whereby DAF-2c signaling may be strengthened to promote taste avoidance behavior.

- 1) AIA responses upon salt concentration changes are enhanced under starvation dependent on the ASER output (Fig 4).
- 2) AIA is a main site of INS-1 release for taste avoidance learning (Supplementary Figs 2, 3).
- 3) Activation of neurons in which INS-1 is ectopically expressed promotes taste avoidance learning via DAF-2c signaling (Fig 3).

We added careful explanation about this point in line 73-77, page 5 in the revised manuscript.

In Fig1b, it is helpful to show the comparison between daf-2c and wt, and between daf-16 and wt. According to the reviewer's suggestion, we added the results of statistical tests comparing the wild type with the *daf-2c* or *daf-16* mutant in Fig. 1b.

In Fig2c, the authors manipulate dopamine signals to interrogate the timing for INS-1 function. A couple of results for high salt conditioning need more clarification. In group 4, the 4th column is significantly different from the 3rd column and in group 5 the 4th column is different from the 1st

(control). Similarly, in Fig3c for high salt conditioning, the 4th column in group 3 shows a significant effect. Some interpretation of these results during the description of the experiments will be very helpful for the clarity of the paper.

We added some interpretation about the results that the reviewer pointed out.

For group 4 in Fig 2c, we speculate that the inhibition of INS-1 release only during a chemotaxis period (3rd column) had a higher inhibition effect on taste avoidance than that during both conditioning and chemotaxis (4th column), based on the finding that INS-1 release during chemotaxis enhances, but that during conditioning inhibits, taste avoidance learning (line 192-194, page13).

For group 5 in Fig 2c, we speculate that prolonged capsaicin exposure had an unknown effect on salt chemotaxis (line 155-159, page10-11).

For group 3 in Fig 3c, we speculate that excess dopamine release itself by the dopaminergic neuron activation might affect salt chemotaxis in a direction that decreases migration toward high salt (line 179-182, page 12).

Do Fig4 and Supplemental Fig4 show the results from the same experiments? If so, it will be much easier to read if the results in Supplemental Fig4 are included in Fig4.

According to the reviewer's suggestion, Supplementary Fig 4 was included in Fig 4.

*Fig5a, it will be more clear to show the comparison between *elg-8* and *wt*, and also between *plc-1* with *wt*.*

According to the reviewer's suggestion, we added the results of statistical tests comparing the wild type with the *elg-8* or *plc-1* mutant in Fig. 5a.

*It is interesting that *elg-8* and *plc-1* have different functions in salt learning, although they both function downstream of *daf-2c* and *akt*. It will be informative if the authors discuss and speculate possible mechanisms for their separate roles.*

We speculate that possible differences in upstream regulators and localization patterns in the plasma membrane might contribute to the separate roles of EGL-8 and PLC-1 in DAF-2c-dependent salt chemotaxis plasticity. It is possible that each of PLCs regulates distinct ASER functions, such as synaptic transmission, at the different sites in the ASER axon. In the future, it will be interesting to clarify how multiple PLCs modulate ASER functions dependent on the axonal DAF-2c signaling in

taste avoidance learning. We added discussion as follows (line 440-464, page 31-32 in the revised manuscript).

“PLC β is known to be directly activated by G α_q ⁴⁰. Also, in *C. elegans*, EGL-30 (G α_q) is proposed to activate EGL-8 in several types of neurons, including ASER²⁵. On the other hand, PLC ϵ is activated by diverse regulators, such as small G proteins⁴⁰. Although regulators for PLC-1 activity are mostly unclear, multiple inputs, including DAF-2c/Akt signaling, may regulate PLC-1 to control taste avoidance after starvation.

In addition to the difference in the activators between PLC β and PLC ϵ , mechanisms for membrane binding and cellular distribution are different. The C-terminal domain (CTD) characteristic for PLC β is known to determine membrane binding and spatial localization to regions of the membrane enriched in the substrate, phosphatidylinositol-4,5-bisphosphate (PIP₂)⁴⁰. Unlike PLC β , the membrane binding regions of PLC ϵ are obscure, although the pleckstrin-homology (PH) domain is proposed to bind PIP₂ and anchors the enzyme at the membrane⁴⁰. In this study, we found the FERM domain in the N-terminal region specific for long isoform of PLC-1 (Fig. 6a). The FERM domain is involved in localization to the plasma membrane and the purified FERM domain binds to PIP₂⁴¹. As a putative Akt-phosphorylation site exists in the FERM domain of PLC-1, DAF-2c/Akt signaling may regulate membrane localization of PLC-1 via phosphorylation of this site, which needs to be confirmed in future studies. The possible differences in upstream regulators and localization patterns in the plasma membrane might contribute to the separate roles of EGL-8 and PLC-1 in DAF-2c-dependent salt chemotaxis plasticity. The ASER neuron has synaptic connections to various interneurons and expresses a variety of neuropeptides, which could extrasynaptically modulate neural circuits. In ASER, each of PLCs may modulate ASER functions, such as synaptic transmission, at the different sites in the axon along with DAF-2c. It will be interesting to elucidate how multiple PLCs modulate ASER functions and its downstream neural circuits to regulate dramatic changes in salt chemotaxis after starvation.”

Response to Reviewer #2 :

We are grateful for the reviewer's comments. It was very helpful to improve our paper.

1. The authors claim that INS-1 secretion drives activity of DAF-2c during the chemotaxis assay to avoid the learned salt concentration. However, according to figure 3d, daf-2c mutants could not be rescued to show the wt phenotype. So, another possibility is therefore that daf-2c mutants did not learn the experience from the first place. Is there another piece of evidence that can reconcile this point? Perhaps the authors could acknowledge and discuss this possibility in the discussion section.

As the reviewer pointed out, we think that DAF-2c may function in conditioning as well as chemotaxis behavior. The axonal transport of DAF-2c is increased during starvation (line 378-379, page 27). This localization change of DAF-2c may be required for acquisition phase of learning. Furthermore, we previously reported that DAF-2 is required for both conditioning and chemotaxis toward odorant using the temperature-sensitive *daf-2(e1370)* mutant (Lin et al., 2010). For the regulation of memory formation during conditioning, we previously suggested that DAF-2a (E11.5-) functions to transmit starvation information, because DAF-2a regulates nuclear translocation of DAF-16 under starvation in the ASER neuron (Nagashima et al., 2019). Based on these findings, DAF-2 plays multiple roles in starvation-dependent learning behavior. As one of those roles, we would like to show the role of INS-1 – DAF-2c signaling during chemotaxis behavior. We added above discussion in line 379-383, page 27 and line 412-421, page 29-30 in the revised manuscript.

2. Similarly, it is claimed that AIA activities are enhanced after starvation and depend on the output from the ASER neuron (lines 213-214). However, synaptic outputs from ASER during learning were inhibited (TeTx). So maybe the animals failed to learn or perhaps the learning process was not effective enough. In this case, the authors could control this by assaying the behavior of ASER TeTx-expressing animals (or chemogenetically inhibit ASER during the learning period only).

According to the reviewer's suggestion, we tested salt chemotaxis of ASER TeTx-expressing worms after feeding or starvation conditioning. As a result, these worms showed a weak but significant defect in taste avoidance after low-salt/starvation conditioning (line 232-233, page 16; Supplementary Fig 4 in the revised manuscript). This data suggests that synaptic output from ASER and possible feedback regulation of INS-1 from AIA may be more important for taste avoidance after low-salt conditioning than that after high-salt conditioning. Consistent of this, we have previously shown that stronger synaptic transmission was observed in ASER after low salt exposure in buffer than that after high salt exposure (Oda et al., 2011). We added above discussion in line 398-404, page 28-29 in the revised manuscript.

Minor:

1. Note some of the refs are clipped (e.g., #22 #39).

We corrected the typo in References.

2. Line 84: should read “animals with a large deletion....”

We corrected as the reviewer pointed out (line 87 in the revised manuscript).

3. Line 99 and line 104: there is a reference to panel d in figure 1, but there is no panel d.

We corrected a reference to Fig 1d to Fig 1c (lines 102, 107 and 276 in the revised manuscript).

4. In legend to figure 1, it will be helpful to add how CI was calculated and what does negative and positive CI mean (it appears in the methods but will be good to mention here as it repeatedly appears throughout the MS).

We added explanation of chemotaxis index in Fig 1, as the reviewer suggested (line 111-115).

5. Figure 2b, will be helpful to add to the legend that continuous line denotes with capsaicin and dotted line without capsaicin.

We added the note, as the reviewer suggested (line 166-167).

Response to Reviewer #3:

We are grateful for the reviewer's comments. It was very helpful to improve our paper.

1). Fig. 1d does not seem to exist. Lines 99, 104 and 244 refer to 1d, which is not displayed.

We corrected a reference to Fig 1d to Fig 1c (lines 102, 107 and 276 in the revised manuscript).

2). For the comparison purpose, it is worth including daf-16 in the high salt conditioning model in the fig 1c, which acts in parallel to daf-2c.

DAF-16 was included in the high salt conditioning model, as the reviewer suggested (Fig 1c).

3). Capital A in fig 1 legend is a typo.

We corrected the typo in Fig 1.

4). Even though it was suggested that AIA is a main site of INS-1 release, the learning defect of ins-1 was not fully rescued by expressing ins-1 in AIA neurons. Instead, ins-1 expression in AWA, ASI or RIC gives a much better rescue (Supplementary fig 2). In supplementary fig 3, it also shows that cell-specific knockdown of AIA does not lead to a dramatic change in taste avoidance learning. This might be caused by inefficient knock-down of ins-1 in AIA by crispr, but it could reflect the contribution for the other neurons. Authors are encouraged to knock-down ins-1 in AWA, ASI and RIC even though its expression in RIC has not been confirmed. The dat-1 single mutant was not investigated in the article, but it has been reported to be defective in salt learning. This might be an indirect evidence that RIC neurons may be involved.

According to the reviewer's suggestion, we examined the effects of INS-1 knockdown in chemosensory neurons, including ASI, and RIC using the *gpc-1* and *tbh-1* promoters, respectively (Supplementary Fig 3). INS-1 knockdown in chemosensory neurons, including ASI, caused significant defects in taste avoidance after both high- and low-salt conditioning, similar to that in AIA. INS-1 knockdown in RIC caused a weak but significant defect in taste avoidance after low-salt conditioning. As the reviewer suggested, INS-1 action in RIC may weakly contribute to the regulation of taste avoidance learning. These results were described in lines 128, 134-136, page 9 in the revised manuscript.

5). Please include the number of animals (n=?) imaged in the fig 4 legend.

The numbers of imaged animals were included in the Fig 4 legend (line 253-256, page 18).

6). egl-8 mutants behave differently in low and high salt migration after high and low salt

conditioning (fig 5a). In low salt migration after high salt conditioning, egl-8; daf-2 double is similar to daf-2 single. However, in high salt migration after low salt conditioning, the double is significantly different from either single. It might not be appropriate to draw one conclusion for two different observations (lines 238 to 241).

As the reviewer pointed out, the *egl-8; daf-2c* double mutant is similar to the *daf-2c* single mutant in low-salt migration after high salt conditioning (Fig 5a). Actually, the *egl-8* mutation caused severe low-mobility phenotypes in salt chemotaxis assay (Supplementary Fig 5), possibly because of the defective function of EGL-8 in motor neurons (Neuron, Vol. 24, 335–346, 1999). Therefore, immobility on chemotaxis test plates was substantially different between *egl-8; daf-2c* double and *daf-2c* single, although chemotaxis index was similar between them. To clarify the effect of decreased EGL-8 activity in the *daf-2c* mutant, we have performed ASER-specific knockdown of *egl-8* in the *daf-2c* mutant, in which mobility is not affected, and have found that *egl-8* knockdown suppressed the low-salt migration defect of *daf-2c* after high-salt conditioning as shown in the original manuscript (Fig 5b left, the fourth block, page 20 in the revised manuscript). To avoid misleading, we edited the description of genetic interaction between *egl-8* and *daf-2c* (line 268-280, page 19) and added the results of low-mobility phenotypes in *egl-8* mutants in line 298-300, page 21 and Supplementary Fig 5 in the revised manuscript.

REVIEWERS' COMMENTS:

Reviewer #1 (Remarks to the Author):

The added discussion and analysis addressed my questions and concerns.

Reviewer #2 (Remarks to the Author):

The authors addressed all my concerns. I support the publication of the revised version.

Alon Zaslaver

Reviewer #3 (Remarks to the Author):

The authors have fully addressed my comments. I do not have any further concerns.